# Nep1-like Proteins from *Valsa mali* Differentially Regulate Pathogen Virulence and Response to Abiotic Stresses

**DOI:** 10.3390/jof7100830

**Published:** 2021-10-04

**Authors:** Jianying Liu, Jiajun Nie, Yali Chang, Lili Huang

**Affiliations:** State Key Laboratory of Crop Stress Biology for Arid Areas, College of Plant Protection, Northwest A&F University, Yangling, Xianyang 712100, Shaanxi, China; liujianying@nwafu.edu.cn (J.L.); niejiajun@nwafu.edu.cn (J.N.); yalinice@163.com (Y.C.)

**Keywords:** *Valsa mali*, Nep1-like proteins, biological roles, function diversification, cytotoxicity, virulence

## Abstract

Necrosis and ethylene-inducing peptide 1(Nep1)-like protein (NLP) is well known for its cytotoxicity and immunogenicity on dicotyledonous, and it has attracted large attention due to its gene expansion and functional diversification in numerous phytopathogens. Here, two NLP family proteins, VmNLP1 and VmNLP2, were identified in the pathogenic fungus *Valsa mali*. We showed that VmNLP2 but not VmNLP1 induced cell death when transiently expressed in *Nicotiana benthamiana*. VmNLP2 was also shown to induce cell death in apple leaves via the treatment of the *Escherichia coli*-produced recombinant protein. *VmNLP1* and *VmNLP2* transcripts were drastically induced at the early stage of *V. mali* infection, whereas only VmNLP2 was shown to be essential for pathogen virulence. We also found that VmNLP1 and VmNLP2 are required for maintaining the integrity of cell membranes, and they differentially contribute to *V. mali* tolerance to salt- and osmo-stresses. Notably, multiple sequence alignment revealed that the second histidine (H) among the conserved heptapeptide (GHRHDWE) of VmNLP2 is mutated to tyrosine (Y). When this tyrosine (Y) was substituted by histidine (H), the variant displayed enhanced cytotoxicity in *N. benthamiana*, as well as enhanced virulence on apple leaves, suggesting that the virulence role of VmNLP2 probably correlates to its cytotoxicity activity. We further showed that the peptide among VmNLP2, called nlp25 (VmNLP2), triggered strong immune response in *Arabidopsis thaliana*. This work demonstrates that NLPs from *V. mali* involve multiple biological roles, and shed new light on how intricately complex the functions of NLP might be.

## 1. Introduction

In plant–pathogen interactions, phytopathogens produce a variety of molecules, such as cell wall-degrading enzymes, effectors, and toxins, in order to establish effective infection on their host plants. Among them, effectors are usually produced to interfere with plant immunity by manipulating plant cellular processes [1,2]. On the contrary, plants can detect effectors, via either cell surface-localized or intracellular immune receptors, to activate immune responses and counteract pathogen infection [3,4]. Remarkably, there are quite a few molecules that can not only promote virulence, but also stimulate immune response on plants, of which, one of the most known examples represents the necrosis- and ethylene-inducing-like proteins (NLPs) [5,6]. 

Originally identified from culture filtrates of *Fusarium oxysporum* [7,8,9], NLPs are a superfamily of effector proteins widely distributed across bacteria, fungi and oomycetes. NLPs can be divided into three types, i.e., type 1, type 2 and type 3, which respectively contain two, four, and six conserved cysteine residues [8,9]. The typical bioactivity of NLPs is to cause cell death in dicotyledons [8,10]. Thus far, NLPs with cell death-inducing activity have been identified in various pathogens. Ottmann and colleagues have proven that NLPs can function as cytolytic toxins that promote the virulence of *Pythium aphanidermatum* [11]. Recently, the receptor for NLPs acting as toxins has been found to be plant sphingolipid glycosylinositol phosphorylceramides (GIPC) [12]. Notably, not all NLPs exhibit cell death-inducing activity in plants, and many characterized NLPs are noncytotoxic. For example, 5 out of 7 NLPs from *Verticillium dahliae*, 11 out of 19 NLPs from *Phytophthora sojae*, and all 10 tested NLPs from *Hyaloperonospora arabidopsidis* are not capable of inducing plant cell death [13,14,15]. Intriguingly, both cytotoxic and noncytotoxic NLPs can serve as immune elicitors. The cytotoxic NLP_Pcc_ from the phytopathogenic bacterium *Pectobacterium carotovorum* can activate the expression of immune marker genes *HIN1* and *PAD3* in tobacco [11]. The 10 noncytotoxic NLPs from *H. arabidopsidis* unexceptionally trigger immune responses when ectopically expressed in *Arabidopsis* [16]. Moreover, a conserved immunogenic peptide among NLPs has been identified, which can be recognized by the receptor-like protein RLP23 in *Arabidopsis* [17,18].

During the past two decades, increasing evidence shows that NLPs play essential roles in diverse plant–pathogen interactions. For instance, VdNLP1 and VdNLP2 from *V. dahliae*, both of which display cytotoxic activity, are induced during pathogen infection and are required for pathogen virulence [15]. VdNLP1 also affects pathogen vegetative growth and conidiospore production [15]. Besides, most *HaNLP* genes from *H. arabidopsidis* are up-regulated at the early infection stages, and the expression is accompanied by appressorium formation, suggesting that these HaNLP proteins may still be functioning in pathogen-plant attachment [13]. In addition, the NLPs from *Phytophthora sojae* are under strong positive selection, implying possible alternative roles in these proteins [14]. In spite of these, the function of noncytotoxic NLPs and cytotoxic NLPs outside pathogen–plant interactions remains largely undiscovered.

Apple *Valsa* canker caused by the ascomycete fungus *Valsa mali*, a weak parasitic pathogenic fungus, annually results in huge economic losses for the apple industry [19,20]. Though much efforts have been made to elucidate the infection mechanism of *V. mali*, the major virulence determinants of this fungus still remain largely puzzled. Much evidence has demonstrated that toxins play important roles during the *V. mali* infection of its apple host [21,22,23,24]. Considering that NLPs frequently function as cytolytic phytotoxins on dicotyledonous plants, we suspect that this member of proteins may play important roles in apple-*V. mali* pathosystem.

In this study, we successfully identified two NLP family proteins in *V. mali*, respectively named VmNLP1 and VmNLP2. We found that VmNLP2 but not VmNLP1 could induce cell death in the model plant *Nicotiana benthamiana*, and VmNLP2 can also cause cell death in the apple host. Both of these two genes were highly induced at early stage of *V. mali* infection. Target gene deletion revealed that only *VmNLP2*-deficient strain exhibited reduced virulence on apple host. Further analysis showed that VmNLP1 and VmNLP2 are differently involved in the maintenance of cell membrane integrity and pathogen tolerance to salt and osmotic stresses. Moreover, the immunogenic peptide among VmNLP2 displayed a potent activity to inducing immune response in *A. thaliana*, whereas the peptide among VmNLP1 showed a much weaker activity. Collectively, our results revealed that VmNLP2 is an essential virulence factor of *V. mali*, and we emphasized that the two NLP proteins from *V. mali* are functionally diversified in multifaceted biological roles, including their cytotoxic activity, contribution to pathogen virulence, and response to abiotic stresses.

## 2. Materials and Methods

### 2.1. Bioinformatics Analysis

The Nep1-like proteins from *Valsa mali* were identified by querying FoNep1 (Accession: AAC97382) from *Fusarium oxysporum* [25,26] against National Center for Biotechnology Information (NCBI) in *V. mali* genome [24]. Signal peptide prediction was performed by SignalP 4.1 [27]. Multi-sequence alignment was performed by Clustal W [28] with Blosum scoring matrix, and the parameters were set as follows: extending gap penalty (0.05), separation gap penalty (0.05), opening gap penalty (10), end gap penalty (10).

### 2.2. Strains and Plant Growth Conditions

The *Valsa mali* wild-type strain 03-8 and *Sclerotinia sclerotiorum* strain 1980 were stored in 20% glycerol at 4 °C. *Escherichia coli* strain DH5α used for plasmid construction was cultured on lysogeny broth (LB) medium at 37 °C, and *Agrobacterium tumefaciens* strain GV3101 used for agroinfiltration of plants was cultured on LB medium at 28 °C. Micropropagated apple (*Malus domestica*) rootstock plants GL-3 and *Arabidopsis thaliana* were cultured in an artificial climate chamber, 12 h/12 h light/dark at 22 °C. *Nicotiana benthamiana* was grown in an artificial greenhouse, 16 h/8 h light/dark at 22 °C.

### 2.3. Plasmid Construction

The open reading frames of *VmNLP1*, *VmNLP2* or *VmNLP2* without signal peptide were amplified from complementary DNA (cDNA) library of *V. mali* with gene-specific primers using Phanta Max Super-Fidelity DNA Polymerase (Vazyme, Nanjing, China). The hemagglutinin (HA)-tagged pCAMBIA1300 vector carrying kanamycin resistance gene was digested with *Kpn*I and was subsequently ligated with the amplicons using ClonExpress II One-Step Cloning Kit (Vazyme, Nanjing, China). For generation of site-mutated proteins, the site-mutated fragments were amplified using mismatched base primers, which were subsequently cloned into pCAMBIA1300 vector using ClonExpress MultiS One-Step Cloning Kit (Vazyme, Nanjing, China). To generate gene complemented constructs, the PDL2 vector was linearized by *Xho*I digestion and further ligated with *VmNLPs* gene fragments that were amplified from *V. mali* genomic DNA containing their native promoters. The primers used for plasmid construction were listed in Appendix A. All PCR products were excised and purified from gel using Gel extraction Kit (OMEGA Biotech, Guangzhou, China). All constructs were sequence-verified by standard vector primers (Sangon Biotech, Shanghai, China).

### 2.4. A. tumefaciens-Mediated Transient Expression and Trypan Blue Staining

Four- to five-week-old *N. benthamiana* plants were used for agroinfiltration assays. The constructs were transformed into *Agrobacterium tumefaciens* GV3101 via electroporation. The colonies were selected by kanamycin (100 µg/mL), rifampicin (100 µg/mL), and confirmed by PCR. The individual correct transformant was cultured in LB medium at 28 °C with 220 rpm for 48 h. Cells were collected by centrifugation and were suspended in MES buffer (10 mM Magnesium chloride, 10 mM 2-(N-morpholino) ethane sulfonic acid (MES), 200 μM acetosyringone, pH 5.7). The suspended *A. tumefaciens* cells were incubated at room temperature for 3 h and were then mixed with P19 silencing suppressor. The final OD_600_ of suspensions were adjusted with MES buffer to 0.6. The suspensions were infiltrated into tobacco leaves using a needleless 1-mL syringe. Symptoms were observed 4–6 d post agroinfiltration. *A. tumefaciens* carrying INF1 and green fluorescent protein (GFP) were used as positive and negative control. Trypan blue staining was performed as described [29]. The experiment was repeated at least three times.

### 2.5. Western Blotting

*N. benthamiana* leaves transiently expressing indicated proteins were collected 36 h after infiltration. Lysing buffer (50 mM Tris, 150 mM NaCl, pH 7.5, 5 mM dithiothreitol (DTT), 0.5% TritonX-100, 1 mM phenylmethanesulfonyl fluoride (PMSF), 1% proteinase inhibitor cocktail (Sigma-Aldrich, St Louis, MO, USA)) was used to extracted total proteins. Ten minutes after centrifuge at 15,000× *g* (4 °C), the supernatant was collected to a new tube. The protein samples were boiled for 6 min in 2× sodium dodecyl sulfate (SDS) loading buffer and were then subjected to sodium dodecyl sulfate polyacrylamide gel electrophoresis (SDS-PAGE). Subsequently, the protein was transferred from gel to a polyvinylidene difluoride (PVDF) membrane under the transfer buffer (20 mM Tris, 150 mM glycine). The membrane was washed by Tris buffered saline (TBS), and TBST (TBS with 0.1% Tween 20) containing 5% nonfat dry milk was used to block the membrane at 50 rpm shaking for 2 h at room temperature. The membrane was then incubated overnight with mouse anti-HA (Abcam, Cambridge, UK, ab18181) monoclonal antibody and rinsed by TBST for three times, followed by staining with goat-anti mouse IgG (Abways, Shanghai, China) secondary antibody. Proteins were detected using ECL subtracted kit (GE Healthcare, RPN2235), as recommended by the manufacturer.

### 2.6. RNA Extraction and qRT-PCR Analysis

Total RNA from collected samples, including leaves of *N. benthamiana* and *A. thaliana*, apple twigs infected with *V. mali*, as well as mycelia from *V. mali*, was extracted using Quick RNA isolation Kit (Huayueyang, Beijing, China) according to the manufacturer’s instruction. RNA concentration and purity were quantified by NanoDrop Micro Photopeter (NanoDrop, Wilmington, DE, USA). The RevertAid First Strand cDNA Synthesis Kit (Thermo Scientific, Waltham, MA, USA) was used for first-strand cDNA synthesis from 1 μg of total RNA, followed by quantitative reverse transcription-PCR (qRT-PCR) using Real Star Green Mixture (GenStar, Beijing, China) with specific primers. *G6PDH* in *V. mali*, *Actin2* in *A. thaliana* or *MdH* in *M. domestica* were used as internal controls. Relative fold changes were calculated using the 2^−ΔΔCT^ method [30].

### 2.7. Generation of Deletion Mutants, Complementation and Overexpression Transformants

To create deletion cassettes, the upstream and downstream flanking sequences amplified from *V. mali* genetic DNA were fused with *NEO* fragments amplified from vector pFL2, using double-joining PCR method [31]. The replacement fragment was transformed into protoplasts of *V. mali* via the polyethylene glycol (PEG)-mediated method [32]. The transformants were selected by G418-resistance followed by four types of PCR verifications to confirm the gene was knocked out. To obtain complementation transformants, pDL2 vector carrying *VmNLPs* or site-directed mutation variants (*VmNLP1*^H127Y^ or *VmNLP2*^Y137H^) were transformed into protoplasts of *VmNLPs* deletion mutants. The hygromycin-resistance transformants were ascertained by PCR with primer pair VmNLP-5F/6R and sequencing. The overexpression transformants were achieved through fusing the fragments with the pDL2 vector containing the *Magnaporthe grisea* ribosomal protein 27 promoter, followed by transformation into the *V. mali* protoplasts. The transformants were filtered by hygromycin and total RNA was isolated to assess transcriptional levels of *VmNLPs* using qRT-PCR analysis. All primers used for gene knockout and overexpression are listed in Appendix A.

### 2.8. Virulence and Phenotypic Analysis of Transformants

For virulence tests, fresh mycelial plugs collected from *V. mali* wild strain (WT), deletion mutants, complementation transformants and overexpression transformants were inoculated on leaves and twigs detached from *Malus domestica* Borkh. cv. Fuji [33]. Five millimeters mycelium plugs used in this assay were taken from the edge of growing colonies cultured on potato dextrose agar (PDA, 20% potato extract (*w/v*), 2% dextrose (*w/v*), 1.5% agar (*w/v*)) medium. Disease lesions were measured 2–3 days post-inoculation (dpi) for apple leaves and 4–5 dpi for apple twigs, followed by a data analysis. For vegetative growth detection, colony diameters of all strains cultured on PDA plates were calculated 2 days post-cultivation at 25 °C. For abiotic stress assays, PDA media were supplemented with congo red (CR, 400 μg/mL), sorbitol (1 M), sodium dodecyl sulfate (SDS, 0.01%) or KCl (0.3 M). Each experiment was conducted with three biological replicates, and all assays were repeated at least three times.

### 2.9. Expression and Purification of Recombinant VmNLP2 Protein

The coding sequence of *VmNLP2* without signal peptide was amplified and cloned into pET28a vector (Novagen Inc., Madison, WI, USA) with *NdeI* and *BamHI* restriction sites, followed by transformation into *E. coli* BL21 (DE3). Cells were cultivated in LB medium at 37 °C until OD_600_ = 0.6~0.8. Subsequently, cells were introduced with 0.3 mM isopropyl-β-D-thiogalactopyrandoside (IPTG) for 24 h at 16 °C, and were then harvested by centrifugation at 5000× *g*. The collected bacteria cells were suspended in lysis buffer (20 mM sodium hydrogen phosphate, 300 mM NaCl, pH 7.4) containing 1 mM phenylmethanesulfonyl fluoride (PMSF), 1.98 mM β-mercaptoethanol and 1 mg ml^−1^ lysozyme followed by sonication. The supernatant of cells was collected via centrifugation at 4 °C and 10,000× *g* for 10 min and were subsequently loaded to Ni-NTA resin (Thermo Scientific, Waltham, MA, USA). VmNLP2 recombinant protein purification was finished by affinity chromatography according to the manufacturer’s instructions. The samples were subjected to silver staining using the Pierce™ Silver Stain for Mass Spectrometry Kit (Thermo Scientific, Waltham, MA, USA) following the manufacturer’s instructions, and were further analyzed by western blotting detection using anti-His monoclonal antibody (Abways, Shanghai, China).

### 2.10. Synthetic Peptides

Flg22 peptide was purchased from Genscript (Genscript Biotech Corporation, Nanjing, China; https://www.genscript.com.cn/) (accessed on 20 November 2020). nlp20 (VmNLP1) and nlp25 (VmNLP2) were synthesized by Sangon Biotech (Sangon Biotech, Shanghai, China; https://www.sangon.com/) (accessed on 17 August 2020). All peptides were dissolved by ultra-pure water to 500 μM for storage, and well diluted to 1 μM in each assay. 

### 2.11. Measurement of Reactive Oxygen Burst

Reactive oxygen species (ROS) generation was detected as described [34]. Briefly, leaf discs were collected from *A. thaliana* seedlings and floated on 0.1 mL ddH_2_O overnight in a 96-well plate. Right before luminescence detection on a Varioskan LUX multimode microplate reader (Thermo Scientific), water was replaced with a 0.1 mL reaction solution (100 μM L-012 (Wako Chemical, Osaka, Japan), 20 μg mL^−1^ peroxidase (Solarbio, Beijing, China)), and 1 μM peptides. Eight biological replicates were used for each sample, and the experiments were repeated at least three times.

### 2.12. Inoculation Assays on A. thaliana

Five- to six-week-old *A. thaliana* seedlings were infiltrated with 1 μM peptides (nlp20 (VmNLP1), nlp25 (VmNLP2) or flg22) or ddH_2_O. Fresh plugs (diameter 5 mm) of *S. sclerotiorum* collected from PDA plates were inoculated on detached *A. thaliana* leaves 24 h post-infiltration. Inoculated leaves were incubated at a high humidity at 24 °C in 1/2 MS (Murashige & Skoog) medium (0.22% (*w*/*v*) MS basal medium (Phytotechnology, Lenexa, KS, USA)). The assay was performed at least three times.

## 3. Results

### 3.1. VmNLP2 but Not VmNLP1 Displays Cytotoxic Activity on N. benthamiana

To identify the NLP gene family in *V. mali*, a blast search of the *V.*
*mali* genome [24] was performed with the FoNep1 from *Fusarium oxysporum* [25,26] in NCBI database. As a result, two different NLP homologues were identified and named VmNLP1 and VmNLP2, respectively (Appendix A). Sequence analysis indicated that both VmNLP1 and VmNLP2 contain a predicted N-terminal signal peptide and two conserved cystine residues (Appendix A), for which they are classified as type 1 NLPs.

To test whether these two proteins exhibit cell death-inducing activity, hemagglutinin (HA)-tagged VmNLP1 and VmNLP2 were transiently expressed in *N. benthamiana* through agroinfiltration, with mCherry-tagged INF1 [35] and HA-tagged GFP used as positive and negative controls, respectively. As shown in Figure 1A, both VmNLP2 and INF1 induced visible cell death 5 days post agroinfiltration (dpa), whereas VmNLP1 and GFP control did not cause any necrosis symptoms. This indicates that VmNLP2 but not VmNLP1 displays cytotoxic activity in *N. benthamiana*. To test whether the signal peptide (SP) is needed for the cytotoxic activity of VmNLP2, VmNLP2 without SP (VmNLP2^ΔSP^) was transiently expressed in *N. benthamiana*. It showed that VmNLP2^ΔSP^ cannot induce necrosis as full length VmNLP2 did (Figure 1A), suggesting that VmNLP2 requires its SP for cell death activation. Western blotting analysis showed that all proteins tested were successfully expressed in *N. benthamiana* (Figure 1B).

### 3.2. VmNLP2 Shows Cytotoxicity on Malus Domestica

To examine whether VmNLP2 displays cytotoxic activity on its natural apple host, we firstly produced VmNLP2 recombinant protein in *E. coli*. N-terminal His-tagged VmNLP2 was verified through silver staining and western blotting detection (Figure 2A). Next, *M. domestica* leaves punctured by sterile syringes at the treatment site were dropped with 5 μM VmNLP2 purified protein, with buffer and 10% methanol used as negative controls, and 40 mM protocatechuic acid [22] used as a positive control. It illustrated that visible necrosis could be observed 24 h after treatment by VmNLP2 purified protein or protocatechuic acid, whereas neither buffer nor 10% methanol control caused any apparent symptoms (Figure 2B). Therefore, VmNLP2 exhibits cytotoxic activity on its apple host as well.

### 3.3. VmNLP2 Is Required for V. mali Virulence

To determine the potential virulence roles of VmNLP1 and VmNLP2, we first analyzed their expression profiles during *V. mali* infection by qRT-PCR. As was shown, both VmNLP1 and VmNLP2 were markedly induced between 6 to 36 h post inoculation (hpi), and both of their transcripts peaked at 6 hpi (Figure 3A). This indicates that VmNLP1 and VmNLP2 are probably involved in the pathogenic process of *V. mali*, particularly during the early stage of pathogen infection.

We next tested whether VmNLP1 and VmNLP2 are required for pathogenicity of *V. mali*, for which, gene knockout mutants (Appendix A) and corresponding complementation transformants (Appendix A) of these two genes were generated via PEG-mediated protoplast transformation. Phenotypic analysis manifested that there was no obvious difference on filamentous growth in all deletion mutants (Appendix A). Inoculation assays on apple host showed that Δ*VmNLP1*-10 and Δ*VmNLP1*-66 were as virulent as the WT strain (Appendix A). Nevertheless, compared to WT strain, the virulence of Δ*VmNLP2*-36 was significantly impaired (Figure 3B). Moreover, by introducing *VmNLP2* into Δ*VmNLP2*-36, the complementation transformant Δ*VmNLP2*-C1 caused comparable disease lesions to the WT strain (Figure 3B). 

In order to further verify the above results, overexpression transformants were generated and their virulence were tested on detached apple leaves. The overexpression transformants of *VmNLP1* (*VmNLP1*-OE6) and *VmNLP2* (*VmNLP2*-OE17) were verified to show six-fold and eight-fold enhanced transcript levels, respectively (Appendix A). Inoculation assays showed that lesion diameter caused by *VmNLP1*-OE6 was not conspicuously different from that caused by WT strain (Appendix A). However, *VmNLP2*-OE17 caused larger disease lesion on apple leaves than that of the WT strain (Figure 3C). We also tested the virulence of these transformants on the detached twigs of apple. As expected, *VmNLP1*-deletion mutants did not result in apparent alterations in virulence compared to the WT strain (Appendix A). Whereas, Δ*VmNLP2*-36 displayed markedly reduced lesion length compared with the WT strain, and the *VmNLP2* complementation transformant was as virulent as the WT strain (Figure 3D). Consistently, the lesion length developed by *VmNLP2*-OE17 was greater than those that of the WT strain (Figure 3D). Taken together, these results demonstrated that *VmNLP2* but not *VmNLP1* is required for *V. mali* full virulence on the apple host.

### 3.4. VmNLP1 and VmNLP2 Differentially Contribute to Salt Tolerance, Osmotic and Membrane Stresses

NLPs have also been demonstrated to occur in both pathogenic and non-pathogenic microorganisms [6,9], and it was assumed that potential function outside plant–pathogen interactions may exist for these proteins [10,15]. Therefore, apart from virulence, whether *VmNLP1* and *VmNLP2* are involved in *V. mali* responsiveness to other environmental factors such as abiotic stresses were subsequently tested. For this, *V. mali* WT strain, gene deletion mutants, and corresponding complementation transformants were cultured in PDA plants supplemented with 0.3 M KCl (salt stress), 1 M Sorbitol (osmotic stress), 0.01% SDS (cell membrane damaging agent), or 400 μg ml^−1^ CR (Congo red, cell wall inhibitor). We found that the vegetative growths of Δ*VmNLP1*-66 and Δ*VmNLP2*-36 were similar to that of the WT under the resistance to cell wall stress caused by CR. Interestingly, Δ*VmNLP1*-66 and Δ*VmNLP2*-36 displayed smaller vegetative growth under the treatment of SDS compared to the WT (Figure 4A), suggesting that both VmNLP1 and VmNLP2 participate in maintaining pathogen cell membrane integrity. In addition, under 1 M Sorbitol, the vegetative growth of Δ*VmNLP1*-66 but not Δ*VmNLP2*-36, was smaller than that of WT (Figure 4A), indicating that VmNLP1 also plays a protective role against osmo-tolerance. Unexpectedly, under 0.3 M KCl, the colony of Δ*VmNLP1*-66 was smaller, whereas Δ*VmNLP2*-36 colony was larger than that of the WT (Figure 4A), indicating that these two NLPs contrastingly regulate *V. mali* tolerance to salt stress. All the complement transformants showed similar tolerance levels to the WT under all tested abiotic stresses (Figure 4A,B). Collectively, *VmNLP1* and *VmNLP2* differentially participate in *V. mali* response to salt, osmotic, and membrane stresses.

### 3.5. The Second Histidine Residue among the Conserved Heptapeptide Is Mutated to Tyrosine in VmNLP2

The difference between VmNLP1 and VmNLP2 in multiple biological roles presented above prompted us to analyze their amino acid sequence. Fifteen fully studied NLPs derived from bacterial, fungal and oomycete species were selected for multiple sequence alignment. It revealed that several residues known to be essential for phytotoxic activity of NLPs, including K116, D117, as well as two cystine residues, are conserved in VmNLP1 and VmNLP2 (Appendix A). The characteristic heptapeptide motif “GHRHDWE” that are considered to be essential for necrosis-inducing activity of NLPs [9,11], are also conversed in almost all selected NLPs (Figure 5). Notably, however, for VmNLP2, the motif is “GHRYDWE”, in which the second histidine (H) among the heptapeptide is mutated to tyrosine (Y) (Figure 5), which may account for the distinct functions between VmNLP1 and VmNLP2.

### 3.6. Site-Directed Mutagenesis of Tyrosine to Histidine Promoted VmNLP2 Cytotoxicity and Virulence

To assay whether the natural residue mutation among the heptapeptide of VmNLP2 affect its the cytotoxicity, constructs with this tyrosine substituted by histidine (VmNLP2^Y137H^) were generated. For comparison, we also replaced the Histidine among VmNLP1 with tyrosine (VmNLP1^H127Y^). When transiently expressed in *N. benthamiana*, neither VmNLP1 nor VmNLP1^H127Y^ was able to induce plant cell death (Figure 6A). Interestingly, however, VmNLP2^Y137H^ apparently exhibited enhanced cytotoxic activity than that of VmNLP2 (Figure 6A). All proteins expressed in *N. benthamiana* were verified by Western blotting (Figure 6B). Thus, this result suggests that the natural residue mutation among VmNLP2 can attenuate its cytotoxicity.

We next tested whether VmNLP1^H127Y^ and VmNLP2^Y137H^ can influence the virulence of *V. mali*. Site-directed mutation transformants Δ*VmNLP1*-C^H127Y^ and Δ*VmNLP2*-C^Y137H^ were generated by introducing VmNLP1^H127Y^ and VmNLP2^Y137H^ fragments into the corresponding deletion mutants. All mutation transformants were ascertained by PCR analysis (Appendix A). Two of them were randomly selected for a virulence test. As was shown, Δ*VmNLP1*-C^H127Y^ caused similar disease lesion as the complementation transformant Δ*VmNLP1*-C1 did, either on the detached apple twigs or leaves (Appendix A). However, compared with Δ*VmNLP2*-C1, Δ*VmNLP2*-C^Y137H^ showed an obvious increment in virulence when tested in detached apple leaves (Figure 6C). To be noted, Δ*VmNLP2*-C^Y137H^ showed no apparent change in virulence on detached twigs (Appendix A), which may be attributed to apple twigs being less sensitive. Altogether, the natural residue mutation among VmNLP2 can also attenuate its virulence, and VmNLP2 contribution to virulence is probably correlated to its cytotoxicity.

### 3.7. nlp25 among VmNLP2 Triggers Strong Immunity Response in A. thaliana

Since both cytotoxic and noncytotoxic NLPs possess a conserved immunogenic peptide [16,18], we next explored whether VmNLP1 and VmNLP2 contain such immunogenic peptides. Therefore, the peptides nlp20 derived from VmNLP1 and nlp25 derived from VmNLP2 (Appendix A) corresponding to nlp20 (NLP_Pp_) [18] were synthesized. The peptides were subsequently used for measurement of immune responses, including ROS burst, expression of immune marker genes, and activation of disease resistance in *Arabidopsis thaliana*, a model plant that recognizes the nlp20 (NLP_Pp_) peptide [17]. The well characterized flg22 peptide [43] was used as a positive control. The luminol-based chemiluminescence assay revealed that both nlp25 (VmNLP2) and nlp20 (VmNLP1) can elicit ROS burst in *A. thaliana* (Figure 7A). Notably, nlp20 (VmNLP1)-induced ROS level was not so strong as that of nlp25 (VmNLP2). We further treated with *A. thaliana* with 1 μΜ peptides and collected the samples 24 h later to analyze the expression of defense genes by qRT-PCR. As was shown in Figure 7B, the transcript levels of *PR1*, *WRKY33* and *FRK1* were markedly induced by treatment of nlp25 (VmNLP2), but not nlp20 (VmNLP1). In accordance, the disease lesions caused by the pathogenic fungus *Sclerotinia sclerotiorum* on *Arabidopsis* pre-treated with nlp25 (VmNLP2) but not nlp20 (VmNLP1) peptide were smaller than that of water control (Figure 7C). These results collectively indicate that nlp25 (VmNLP2) possesses a strong immunogenic activity, whereas nlp20 (VmNLP1) shows a much weaker and even no apparent immunogenic activity in *A. thaliana*.

We also tested whether the synthetic peptides exhibited immunogenic activities in *N. benthamiana*. As a result, none of them triggered apparent ROS bust (Appendix A), which is consistent with the fact that *N. benthamiana* lacks the RLP23 receptor for detection of nlp20 [17,18]. Most recently, two NLPs from *Pythium oligandrum* have been demonstrated to activate immune responses in *N. benthamiana* [44]. We next tested whether VmNLP recombinant proteins function similarly. As was shown, neither of them obviously induced ROS burst (Appendix A), nor did they activate the expression of immune marker genes like *NbPR1*, *NbPR2* and *NbPR4* in *N. benthamiana* (Appendix A). Thus, VmNLPs and the synthetic peptides therein display no immunogenic activity in the model plant *N. benthamiana*.

## 4. Discussion

The function of NLPs is varied due to the broad taxonomic distribution and rapid evolution [8,14]. In this study, we found that two NLPs from *Valsa mali* participate in a variety of biological roles and exhibit functional diversification. We have shown that only one of the two NLP proteins in *V. mali*, VmNLP2, possesses cytotoxic activity and contributes to virulence of *V. mali*. The non-cytotoxic VmNLP1 does not contribute to pathogenicity. However, we found that VmNLP1 positively regulates *V. mali* tolerance to salt, osmotic stress and cell membrane damaging agent. VmNLP2 was also shown to be involved in maintaining the integrity of cell membrane, but VmNLP1 and VmNLP2 played a distinct role in response to salt stress. Moreover, we showed that the immunogenic peptide nlp25 (VmNLP2) triggered strong immune responses in *Arabidopsis thaliana*, whereas nlp20 (VmNLP1) only elicited much weaker immune responses in *A. thaliana*.

Based on the ability to induce necrosis, NLPs can be divided into cytotoxic and non-cytotoxic NLPs. The cytotoxic NLPs can permeabilize plant plasma membranes of dicotyledons and lead to necrosis [7,11]. The non-cytotoxic NLPs are not capable of inducing cell necrosis, and their biological functions are largely mysterious [13,14]. Most necrotrophic phytopathogens have only one or two NLPs, and a majority of them, if not all, have been demonstrated to be cytotoxic. For example, *Botrytis elliptica* contains two NLPs which exhibit cytotoxic activity in many tested dicots [45], and two NLPs from *Sclerotinia sclerotiorum* can also induce necrosis when transiently expressed in tobacco leaves [36]. Here, we found that VmNLP2 but not VmNLP1 from the weakly parasitic *V. mali* exhibited cytotoxic activity in plants (Figure 1 and Figure 2). Actually, there are also some fungal species that contain necrotrophic phase encoding noncytotoxic NLPs on account of gene expansion, such as *Verticillium dahlia* [15] and *Colletotrichum higginsianum* [46]. It is worth noting that the mature protein of *Botryis cinerea* NLPs fused with the signal peptide of tobacco PR1a can trigger necrotic responses in *N. benthamiana*, whereas the BcNEP1 and BcNEP2 carrying their endogenous SP did not cause necrosis [37]. Likewise, it is possible that the inability of VmNLP1 to induce plant cell death may attribute to the potential inefficiency of its endogenous SP. Alternatively, the tertiary structure of VmNLP1 may not be conductive to its toxic function, similar to the case of HaNLP3, a typical noncytotoxic NLP from *Hyaloperonospora arabidopsidis* [13]. 

The transcripts of *VmNLP1* and *VmNLP2* were found to be highly induced at early stage of infection (Figure 3A), and we therefore hypothesized that these two genes may be involved in pathogenicity. Deletion of *VmNLP2* reduced virulence on apple leaves and twigs (Figure 3B,D), however, the knocking out of *VmNLP1* posed no apparent effect on pathogen virulence (Appendix A). In accordance, the overexpression of *VmNLP2* in *V. mali* promoted pathogen infection in apple leaves (Figure 3C). This evidence together demonstrated that VmNLP2 is an important virulence factor of *V. mali*. This is similar to several characterized NLPs from other phytopathogens, which serve as virulence factors as well. For example, targeted deletion *nip* gene in *E. carotovora* subsp. *carotovora* compromised virulence in potato [42]; *V. dahliae* lacking either of the two cytotoxic NLPs showed significantly reduced virulence on tomato as well as on *Arabidopsis* plants [15]; overexpression of FoNep1 in the fungus *Colletotrichum coccodes* improved fungal virulence on *Abutilon theophrasti* [47]. Importantly, by multiple sequence alignment, we observed a residue mutation among the conserved heptapeptide motif of VmNLP2, in which the second histidine was mutated to tyrosine (Figure 5). We further found that the cytotoxic activity and virulence of VmNLP2 were simultaneously enhanced when the tyrosine residue was replaced with histidine residue (Figure 6), indicating a potential positive correlation between VmNLP2 cytotoxicity and its contribution to virulence. This result is reminiscent of NLP_Pcc_ from *Pectobacterium carotovorum*. It has been reported that the complementation of less cytotoxic NLP_Pcc_ variants in NLP_Pcc_-deficient *P. carotovorum* strain resulted in correspondingly declined virulence [11]. These findings reinforce that NLPs can act as toxin-like virulence factors during the interaction of plant and pathogens.

Previous studies reveled that the large expansion of Nep1-like proteins family not only in oomycetes but also in fungi [6]. Thus, we were also interested in exploring the biological roles of VmNLPs apart from virulence and cytotoxic activity. In this study, we found that both VmNLP1 and VmNLP2 are involved in the salt tolerance and maintenance of cell membrane integrity of *V. mali* (Figure 4). In addition, VmNLP1 also participated in response to osmotic stress (Figure 4). Meanwhile, considering that NLPs are secreted proteins, we cannot exclude that VmNLP1 and VmNLP2 serve as ‘guards’ to differentially protect *V. mali* against abiotic stresses. A role of NLP in growth and conidiospore production has also been confirmed in the fungus *V. dahlia* [15]. Furthermore, the high expression of *HaNLPs* at an early infection stage led researchers to hypothesize it may play a role in zoospore attachment or primary contact between pathogen and host [13]. Our results here indicate that the two NLPs from *V. mali* are functionally diversified, not only in cytotoxicity and virulence, but also in regulation of pathogen responses to abiotic stresses. 

In this study, we found that both the peptide nlp25 (VmNLP2) and nlp20 (VmNLP1) can elicit ROS burst in *A. thaliana*, however, ROS-triggered by nlp20 (VmNLP1) was much lower than that of nlp25 (VmNLP2) (Figure 7A). Additionally, nlp25 (VmNLP2) but not nlp20 (VmNLP1) activated the expression of defense-related genes and induced *Arabidopsis* resistance against *Sclerotinia sclerotiorum* (Figure 7B,C), though nlp20 (VmNLP1) is more similar to nlp20 derived from NLP_Pp_ [18]. This result further highlights the functional diversification of VmNLPs. We have also tested the response of apple leaves to these two peptides. Nevertheless, both nlp20 (VmNLP1) and nlp25 (VmNLP2) failed to trigger the expression of defense-related genes on apple, nor could they promote apple resistance to the *V. mali* infection (Appendix A). It is possible that apple lacks a receptor like RLP23 from *Arabidopsis* [17] for detection of the immunogenic peptide among NLPs. 

## 5. Conclusions

In summary, our study revealed that VmNLP2 is a cytotoxic NLP and it serves as an essential virulence factor of *V. mali*. Surprisingly, we showed that VmNLP1 and VmNLP2 play contrasting roles in *V. mali* response to cell membrane damaging agent, salt- and osmic-tolerance. The two NLPs in *V. mali* play functionally diversified roles in cytotoxicity, virulence, and regulation of pathogen tolerance to abiotic stresses, which offer a new understanding of the biological roles of the Nep1-like proteins.

## Figures and Tables

**Figure 1 jof-07-00830-f001:**
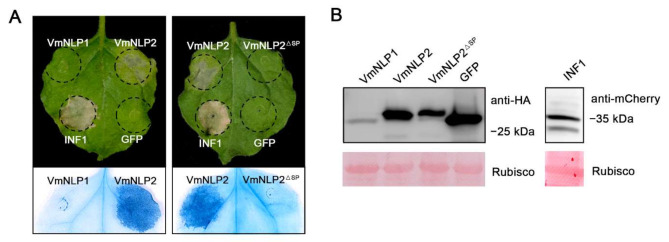
Transient expression of VmNLP2 induces cell death in *N. benthamiana*. (**A**) Representative *N. benthamiana* leaves showing cell death triggered by expressed proteins. VmNLP1, VmNLP2, VmNLP2 without signal peptide (VmNLP2^ΔSP^), GFP, and INF1 were transiently expressed in *N. benthamiana*. The leaves were photographed 5 d post agroinfiltration. Trypan blue staining was used to visualize the symptoms of cell death. (**B**) Immunoblot analysis on total proteins extracted from *N. benthamiana* with anti-HA or anti-mCherry antibodies. Rubisco protein was stained with Ponceau S as a loading control.

**Figure 2 jof-07-00830-f002:**
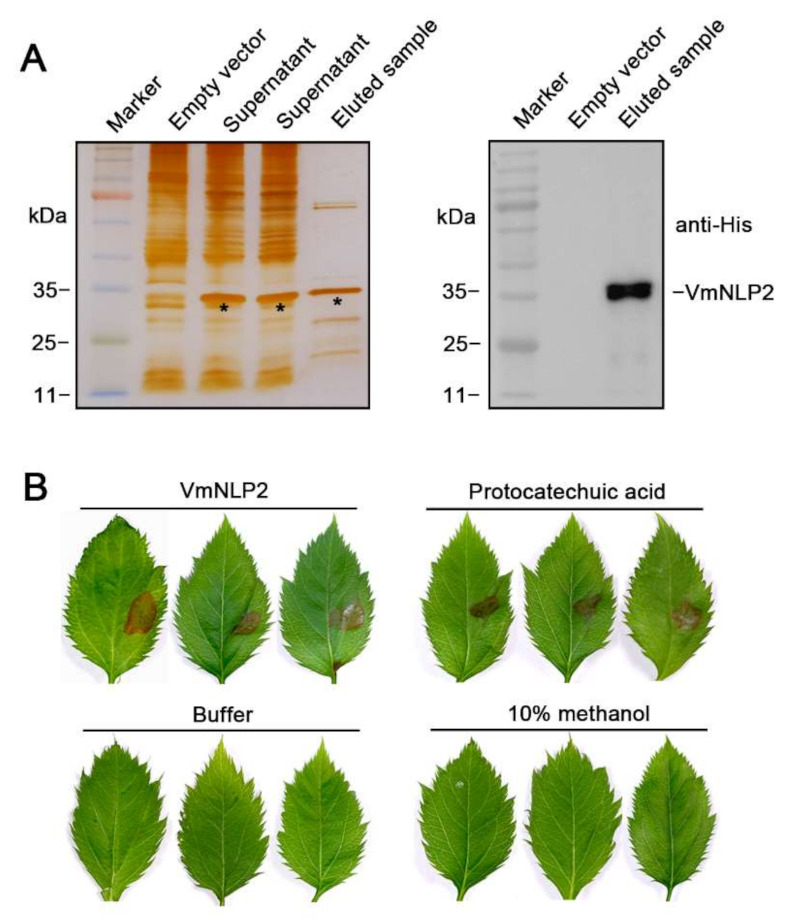
VmNLP2 recombinant protein displays cytotoxic activity on apple leaves. (**A**) Induced expression of VmNLP2 recombinant protein in *E. coli*. Silver-stained SDS-PAGE (left panel) and immunoblot analysis with anti-His antibody (right panel) were performed to detect VmNLP2 recombinant protein. Protein bands of interest were indicated by black asterisks. (**B**) Representative apple leaves showing cell death triggered by VmNLP2 purified protein. 5 μM VmNLP2 purified protein, buffer, 10% methanol and 40 mM protocatechuic acid were dropped on the needle-pricked area of apple leaves. Pictures were taken 2 days after treatment.

**Figure 3 jof-07-00830-f003:**
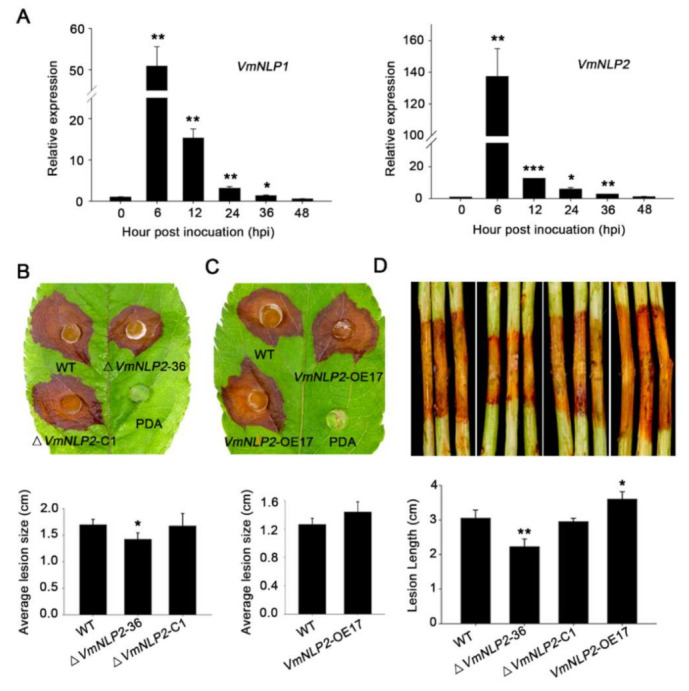
VmNLP2 is important for *V. mali* pathogenesis during infection on apple. (**A**) Expression profiles of *VmNLP1* and *VmNLP2* during *V. mali* infection of apple. Relative transcript levels of *VmNLP1* and *VmNLP2* at 0, 6, 12, 24, 36, and 48 hpi were determined by qRT-PCR. *G6PDH* gene in *V. mali* was used as an internal reference. Similar results were obtained in three independent experiments. Bars indicate standard deviations (SD) from three technical replicates. Asterisks indicate significant differences with 0 h based on Student’s *t*-test (*, *p* ≤ 0.05; **, *p* ≤ 0.01; ***, *p* ≤ 0.001). (**B**) Virulence tests of WT, *VmNLP2* deletion mutant (Δ*VmNLP2*-36) and complementation transformant (Δ*VmNLP2*-C1). (**C**) Virulence tests of WT and *VmNLP2* overexpression transformant (*VmNLP2*-OE17) on detached apple leaves. The experiment was repeated four times with six leaves per biological replicate. Representative photographs were taken at 2 dpi. Average lesion sizes (lesion diameter) were measured and the statistical analyses were performed by Student’s *t*-test (*, *p* ≤ 0.05). Bars indicate ±SD. (**D**) Virulence tests of WT, Δ*VmNLP2*-36, Δ*VmNLP2*-C1 and *VmNLP2*-OE17 on detached apple twigs. Similar results were obtained from at least three independent experiments. Representative photographs were taken 4 dpi. Lesion lengths were measured and the statistical analyses were performed by Student’s *t*-test (*, *p* ≤ 0.05; **, *p* ≤ 0.01). Bars indicate ±SD.

**Figure 4 jof-07-00830-f004:**
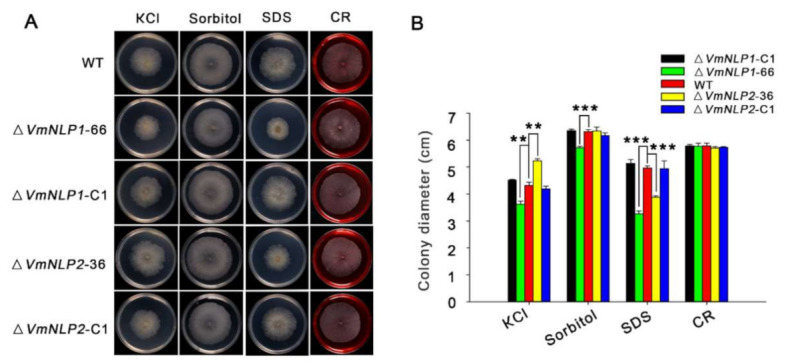
VmNLP1 and VmNLP2 differentially contribute to *V. mali* tolerance to salt, osmotic and membrane stresses. (**A**) Representative photographs showing the growth of all strains under abiotic stresses. WT, Δ*VmNLP1*-66, Δ*VmNLP1*-C1, Δ*VmNLP2*-36 and Δ*VmNLP2*-C1 were cultured on PDA medium and PDA supplemented with 0.3 M KCl, 1 M sorbitol, 0.01% sodium dodecyl sulfate (SDS) and 400 μg/mL Congo red (CR) at 25 °C for 3 days, respectively. The experiment was performed in triplicate, with five petri dishes in each repetition. (**B**) Colony diameter calculation for growth of the stains 3 days after inoculation on tested medium. Statistical analyses were performed by Student’s *t*-test (**, *p* ≤ 0.01; ***, *p* ≤ 0.001). Bars indicate ± SD of the mean of three replicates.

**Figure 5 jof-07-00830-f005:**
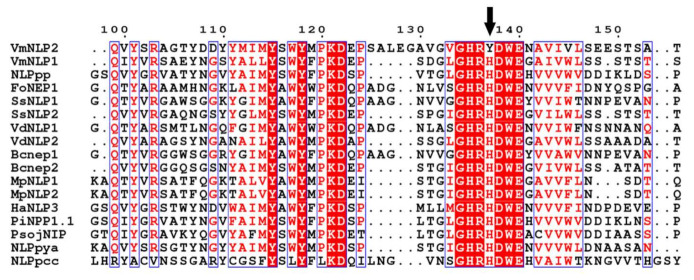
The second histidine residue among the conserved heptapeptide of VmNLP2 is mutated to tyrosine. A total of 17 NLPs, including VmNLP1 and VmNLP2 from *Valsa mali*, NLP_pp_ from *Parasitic phytophthora* [18], FoNep1 from *Fusarium oxysporum* [25], SsNLP1 and SsNLP2 from *Sclerotinia sclerotiorum* [36], VdNLP1 and VdNLP2 from *Verticillium dahlia* [15], Bcnep1 and Bcnep2 from *Botryis cinerea* [37], MpNLP1 and MpNLP2 from *Moniliophthora perniciosa* [38], HaNLP3 from *Hyaloperonospora arabidopsidis* [13], PiNPP1.1 from *Phytophthora infestans* [39], PsojNIP from *Phytophthora sojae* [40], NLP_pya_ from *Pythium aphanidermatum* [41], and NLP_pcc_ from *Pectobacterium carotovorum* [42] were used for multiple sequence alignment by Clustal W. Part of the alignment results are shown. Conserved residues are shaded on a red background. Amino acid mutation among conserved heptapeptide is indicated by a black arrow.

**Figure 6 jof-07-00830-f006:**
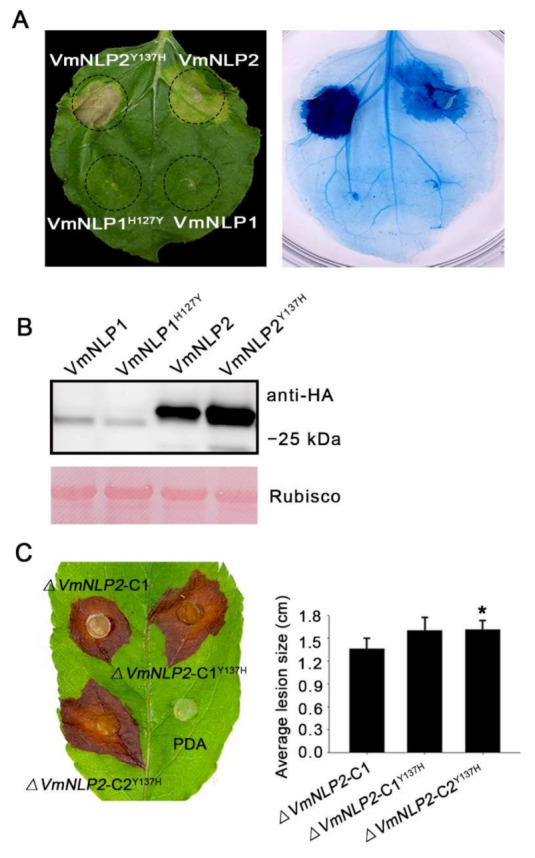
Substitution of the tyrosine with histidine promotes VmNLP2 cytotoxicity and contribution to virulence. (**A**) Representative *N. benthamiana* leaves showing cell death triggered by VmNLP2 and the mutation variant VmNLP2^Y137H^. VmNLP1, VmNLP2, VmNLP1 with introduced tyrosine substitution (VmNLP1^H127Y^) and VmNLP2 with introduced histidine substitution (VmNLP2^Y137H^) were transiently expressed in leaves. The cell death symptoms were assessed and photographed 4 days post-infiltration. Trypan blue staining was used to further visualize plant cell death. (**B**) Immunoblot analysis of proteins extracted from *N. benthamiana* transiently expressing indicated proteins. Rubisco protein was stained with Ponceau S as a loading control. (**C**) Virulence tests of complementation transformant (Δ*VmNLP2*-C1), or complementation transformant with introduced histidine substitution (Δ*VmNLP2*-C1^Y137H^, Δ*VmNLP2*-C2^Y137H^) on detached apple leaves. Similar results were obtained from three independent experiments. Photographs were taken at 2 dpi. Average lesions were calculated and the statistical analyses were performed by Student’s *t*-test (*, *p* ≤ 0.05). Bars indicate ± SD.

**Figure 7 jof-07-00830-f007:**
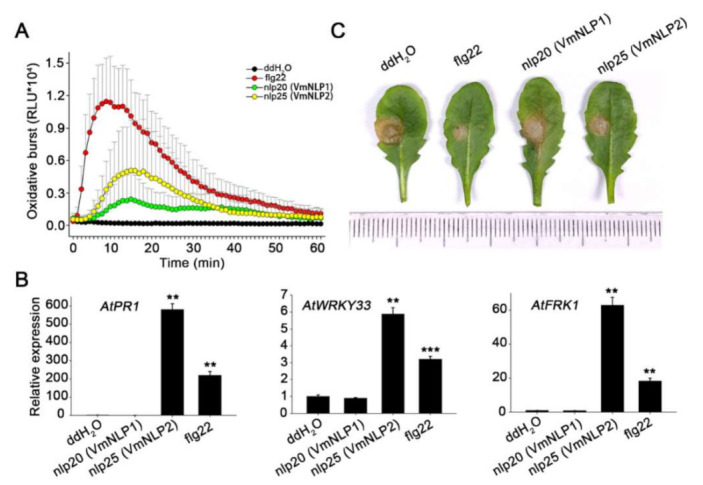
nlp25 peptide derived from VmNLP2 triggers plant immunity responses in *A. thaliana*. (**A**) Oxidative burst in *A. thaliana* treated with 1 μΜ peptides (nlp20 (VmNLP1), nlp25 (VmNLP2), flg22) or ddH_2_O. Values are showed as relative light units (RLU). (**B**) VmNLP2 activated transcript accumulation of defense related genes in *A. thaliana*. The leaves were infiltrated with 1 μΜ peptides (nlp20 (VmNLP1), nlp25 (VmNLP2), flg22) or ddH_2_O control. Gene expression was assessed 24 hpi by qRT-PCR. *Actin2* was used as an endogenous reference. Means and SDs were calculated from three independent biological replicates. Significant differences based on student’s *t*-test are indicated by the asterisks (**, *p* ≤ 0.01; ***, *p* ≤ 0.001). (**C**) nlp25 (VmNLP2) promoted *A. thaliana* resistance against *S. sclerotiorum*. The leaves were infiltrated with 1μΜ synthetic peptide nlp20 (VmNLP1) or nlp25 (VmNLP2) 24 h before inoculation of *S. sclerotiorum*. The flg22 peptide and ddH_2_O were used as positive and negative control, respectively. The experiments were performed three times with similar results.

## Data Availability

All data are available within the article and Appendix A.

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
