# Peer review of "Nep1-like Proteins from Valsa mali Differentially Regulate Pathogen Virulence and Response to Abiotic Stresses"

_jof, 2021, doi:10.3390/jof7100830_

Round 1

Reviewer 1 Report

Reviewer’s comments

In this study, the authors explored the functional roles of Nep1-like proteins (NLPs) from Valsa mali through different approaches. First, the authors demonstrated a cytotoxic effect of VmNLP2 through agrofiltration on Nicotiana benthamiana and recombinant protein treatment on Malus domestica leave,. Genetic manipulation of V. mali revealed that VmNLP2 serves as a virulence factor for infection on apple leaves and twigs. In addition, VmNLPs may have a potential role in abiotic stress response in culture. Bioinformatic analyses and site-directed mutagenesis showed that a natural mutation from a conserved histidine residue to tyrosine (H127Y) of VmNLP2 contributes to reduced cytotoxic effect and virulence activity. Finally, the authors found an immunogenic role of synthetic VmNLP2 in Arabidopsis thaliana through hypersensitive response and systemic-acquired resistance.

Overall, I appreciated several experiments conducted by the authors to demonstrate various functional roles of VmNLPs. However, some observations do not have supporting data on underlying mechanisms. Unless having more supporting evidence, the authors may need to carefully interpret results and make conclusions based on the data they have. Writing organization, language editing and proofreading are required for further consideration. Please see below for my comments:

Major comments

- I am skeptical about how the authors used the words ‘cytotoxic’ and ‘immunogenic’. The authors showed a cytotoxic effect of VmNLP2 through agrofiltration on Nicotiana benthamiana and recombinant protein treatment on Malus domestica leaves. For treatment of synthetic peptides on A. thaliana, the leaves have hypersensitive response (ROS burst), which leads to cell death too. Considering this, VmNLP2 can be counted as having a cytotoxic effect on A. thaliana. However, the authors concluded that VmNLP2 has an immunogenic effect on A. thaliana. In my opinion, cell death in A. thaliana should not be counted as cytotoxic as it is a programmed cell death to induce an immune response (systemic acquired resistance), but whether VmNLP2 also has an immunogenic activity to other hosts need to be more investigated. Have the authors conducted the ROS burst assay on N. benthamiana and M. domesticus to prove whether VmNLP2 leads to hypersensitive response, or just necrosis? Testing genetic markers for cell death and immune response, especially for N. benthamiana, are also important to rule out the immunogenic activity in these two hosts. If the authors could not provide further proofs, I recommend the authors to avoid using the word ‘cytotoxic’ as an underlying mechanism of cell death is not clearly illustrated. Also, the authors may want to clarify if the immunogenic activity of VmNLP2 on N. benthamiana and M. domestica has been thoroughly investigated.

- I wonder why the authors utilize several approaches for VmNLP2 treatment: agrofiltration in N. benthamiana, treating a recombinant protein in apples, and treating a synthetic peptide in A. thaliana. Have the authors confirmed that different treatment approaches do not cause a difference in occurred symptoms? For example, have the authors tried three treatment approaches in A. thaliana to make sure there is no difference in occurred symptoms?

- Based on Supplementary Figure 8, I wonder if the transcript level of MdPR1 and MdPDF1.2 in the nlp25 treatment is lower than other treatments. If they are statistically lower, I think this result looks very interesting to show that VmNLP2 can function as an immunogen in one host (A. thaliana) but as an effector/virulent factor in the other host (M. domesticus). The authors could conduct a statistical test to confirm this.

- I am personally not convinced that VmNLPs directly contribute to response to abiotic stresses. For example, sodium dodecyl sulfate (SDS) has been known as a protein denaturant and a membrane perturbing agent. VmNLPs secreted to extracellular fraction may interact with SDS to hamper their toxicity effect on fungal cells, and that both VmNLP knockout mutants have fewer growth due to fewer secreted proteins to neutralize SDS. Secreted proteins can also affect osmolarity of surroundings. Thus, having less secreted proteins through gene knockout can alter osmolarity, which affect fungal growth. Thus, the data found by the authors can be just due to an amount of secreted proteins. The authors need to provide more evidence to prove that their observation is due to a biological role of VmNLP. For instance, have the authors checked that the expression of VmNLP in abiotic stress conditions is more than a normal growth condition in wild-type strain? Did the authors have other markers to prove a response of abiotic stresses? If the authors could not provide further evidence, I recommend saying the VmNLP may have ‘a protective role against abiotic stress’ instead of ‘a response to abiotic stress’.

- In Figure 4, I do not think the treatment with H2O2 would provide any useful information. If the authors add 6 mM of H2O2 directly on the plate before fungal inoculation, H2O2 is highly likely to be degraded before fungi start growing.

- Please provide details in figure captions, and explanation of details in supplementary figures. It is hard for readers to understand the content by looking at graphics alone.

Minor comments

- Please arrange supplementary figures in chronological order.

- Line 49: Please change to ‘many characterized NLPs are noncytotoxic’.

- Line 62: Please change to ‘vegetative growth and conidiospore production’

- Line 63: Please specify HaNLP. Which species does it come from?

- Line 66: Should this be ‘under’, instead of ‘underling’?

- Line 67: Please change to ‘implying possible alternative roles’

- Line 70: Please change to ‘the ascomycete fungus’; ‘weak pathogenic fungus’

- Line 75: Please change to ‘on dicotyledonous plants’

- Line 94: Please indicate which species. Fusarium oxysporum? It would be useful to indicate accession number of the reference sequence.

- Line 95: Has V. mali genome been published before? If so, please cite a publication of the genome. Otherwise, please indicate if it is unpublished data from someone.

- Line 96ff: Please provide a suitable citation for SignalP 4.1:

Nielsen, Henrik. "Predicting secretory proteins with SignalP." Protein function prediction. Humana Press, New York, NY, 2017. 59-73.

- Line 97: Which platform did the authors perform Clustal W alignment? Please also provide a proper citation of the platform the authors used.

- Line 101ff: Are these strains from public culture repositories? If so, please indicate where these cultures come from. If not, I recommend the authors to deposit these strains in public culture collection.

- Line 102: Please use a standard chemical name ‘glycerol’, instead of ‘glycerin’.

- Line 103: Please verify if LB stands for ‘Luria-Bertani’ or ‘Lysogeny broth’. Please also italicize Agrobacterium tumefacians.

- Line 114: Please provide details how the authors generated site-mutated cDNA fragments. Probably the authors used mismatch base primers, didn’t they?

- Line 133: Please clarify the induction buffer. Does it have a known formula? Or is it retrieved from a company kit?

- Line 155: Total RNA from what? Apple leaf tissues infested with V. mali? Please clarify.

- Line 182ff: Please use the phrase ‘Five millimeters’. Did the authors just place agar plugs directly on leaves as an inoculation assay? If so, please specify.

- Line 184: Please indicate if the percentage is weight by volume, or % (w/v).

- Line 186: Should this be ‘lesion size’? Please clarify.

- Line 188: Please subscript H2O2.

- Lines 188, 348: Shouldn’t SDS stand for ‘sodium dodecyl sulfate’? Please verify.

- Line 199: Please use the symbol ‘b-mercaptoethanol’.

- Line 211: Dissolved at what concentration?

- Line 233ff: The authors should show the alignment with other NLPs and highlight conserved cysteine residues to verify that these VmNLP1 and VmNLP2 are type 1 NLP. Probably the authors can refer to Supplementary Figure 5.

- Line 243ff: Did the authors ensure that it is necrosis? Can it be hypersensitive response?

- Line 249: Should this be ‘VmNLP2 without signal peptide (VmNLP2DSP)’?

- Figure 1B, 2B, 6B: Please change a label be ‘anti-His’ instead of ‘a-His’, as well as ‘anti-mCherry’ instead of ‘a-mCherry’. Using the a symbol may confuse readers that it is a subunit of protein.

- Figure S1: I do not understand the figure. Please provide a figure caption and details for each lane. For example, in S1A panel, what is the DVmNLP1-10? A deletion mutant? Why does it have four different PCR lanes? In S2B panel, please specify that ‘-C1’, ‘-C2’ and ‘-C3’ are different complementation lines.

- Line 285: Virulence assay on what host? Apple? Please clarify.

- Line 325ff: Please clarify the rationale of testing this.

- Lines 362, 374, 377, 378, 402: Please change ‘histone’ to ‘histidine’.

- Line 472ff: Have the authors tried running protein structure modeling to see if VmNLP1 structure is different from other NLPs? Judging from the alignment, I am not sure if its tertiary structure is different from other NLPs.

- Line 517ff: Have the authors checked if there is a reference genome of apple available, and check if RLP23 is present in the genome? What about N. benthamiana?

Reviewer 2 Report

The paper by Liu and coworkers describe the role of two NLP proteins from Valsa mail and their involvement in plant-pathogen response in the model plant Nicotiana benthemaniana and Malus domestica. Moreover, these proteins, namely VmNLP1 and VmNLP2, were functionally characterized with the aim to assess differences in terms to respond to abiotic stress and virulence level. The research work is well written and designed, and the performed experiments allowed to drawn important conclusions related to the functional role of the aforementioned proteins. The paper is well written and the data reported are solid. No plagiarism was detected. In this view, I consider the paper adequate for publication in the Journal of Fungi. I just recommend fixing the minor issues reported below.

- line 140, please insert a space in "pH7.5"
- line 145, please check the verb "transformed"
- line 167, please correct the term "resisrance"
- line 188, please correct H2O2
-line 218, please insert a space before "peroxidase"

Author Response

The paper by Liu and coworkers describe the role of two NLP proteins from Valsa mail and their involvement in plant-pathogen response in the model plant Nicotiana benthemaniana and Malus domestica. Moreover, these proteins, namely VmNLP1 and VmNLP2, were functionally characterized with the aim to assess differences in terms to respond to abiotic stress and virulence level. The research work is well written and designed, and the performed experiments allowed to drawn important conclusions related to the functional role of the aforementioned proteins. The paper is well written and the data reported are solid. No plagiarism was detected. In this view, I consider the paper adequate for publication in the Journal of Fungi. I just recommend fixing the minor issues reported below.

- line 140, please insert a space in "pH7.5"
- line 145, please check the verb "transformed"
- line 167, please correct the term "resisrance"
- line 188, please correct H2O2
-line 218, please insert a space before "peroxidase"

Thanks very much for taking your time to review our manuscript. We really appreciate all your comments and suggestions. The manuscript has been revised extensively according to your kind suggestions. We have carefully scrutinized the manuscript and made corresponding revisions including some typos and grammatical errors. Please find my response in below and my revisions in the resubmitted files.

Point 1: line 140, please insert a space in "pH7.5"

Response 1: Revised.

Point 2: line 145, please check the verb "transformed"

Response 2: We are very sorry for our incorrect writing. We’ve changed "transformed" to "transferred" in the revised version.

Point 3: line 167, please correct the term "resisrance"

Response 3:  The typo is revised in our resubmitted manuscript. Thanks for your correction.

Point 4: line 188, please correct H2O2

Response 4: Revised.

Point 5: line 218, please insert a space before "peroxidase"

Response 5: Thanks for your careful checks. We have corrected the errors accordingly.

Round 2

Reviewer 1 Report

Comments for authors

Thank you very much for considering my comments to improve the manuscript. I appreciate all efforts the authors have made to address my concerns, and I am truly satisfied for the authors’ responses. I highly recommend this manuscript for publication. Please see a few final comments from me:

- The assay that tested immunogenicity of NLPs in N. benthamiana is nicely presented in the author response. I think it would be awesome if the authors could provide this as another supplementary figure. The authors have put a great effort to prove my concern, and there is no point that the authors exclude this from the manuscript.

- Supplementary figure 8: Did the authors have a positive control that shows oxidative burst in M. domestica? I do not know if flg22 can trigger oxidative burst in apple leaves, but it would be great if having a positive control to compare.

- Please again double check the spellings of all technical terms and chemical names. For example, please change ‘vegetable growth’ to ‘vegetative growth’ in lines 188 and 192.

Author Response

Thank you very much for considering my comments to improve the manuscript. I appreciate all efforts the authors have made to address my concerns, and I am truly satisfied for the authors’ responses. I highly recommend this manuscript for publication. Please see a few final comments from me:

- The assay that tested immunogenicity of NLPs in N. benthamiana is nicely presented in the author response. I think it would be awesome if the authors could provide this as another supplementary figure. The authors have put a great effort to prove my concern, and there is no point that the authors exclude this from the manuscript.

- Supplementary figure 8: Did the authors have a positive control that shows oxidative burst in M. domestica? I do not know if flg22 can trigger oxidative burst in apple leaves, but it would be great if having a positive control to compare.

- Please again double check the spellings of all technical terms and chemical names. For example, please change ‘vegetable growth’ to ‘vegetative growth’ in lines 188 and 192.

We are grateful to you for reviewing the manuscript carefully and providing all the constructive suggestions. The manuscript has been extensively revised once again based on your valuable suggestions. Here are responses to your comments:

Point 1: - The assay that tested immunogenicity of NLPs in N. benthamiana is nicely presented in the author response. I think it would be awesome if the authors could provide this as another supplementary figure. The authors have put a great effort to prove my concern, and there is no point that the authors exclude this from the manuscript.

Response 1: Many thanks for your suggestion. Accordingly, the result for immunogenicity test of NLPs in N. benthamiana has been included in the resubmitted manuscript, as seen in the lines 444-453 and the new Supplementary Figure S8.

Point 2: - Supplementary figure 8: Did the authors have a positive control that shows oxidative burst in M. domestica? I do not know if flg22 can trigger oxidative burst in apple leaves, but it would be great if having a positive control to compare.

Response 2: Thank you for pointing this out. We actually used flg22 as a potential control in our assay designs. However, flg22 failed to trigger ROS burst in apple leaves. We also failed to find any positive controls in this assay. For this, only negative control(s)  was used.

Point 3: - Please again double check the spellings of all technical terms and chemical names. For example, please change ‘vegetable growth’ to ‘vegetative growth’ in lines 188 and 192.

Response 3:  We have carefully checked the entire manuscript for spelling errors again. The typos were revised in our resubmitted manuscript.